# Thrombocytes and Platelet-Rich Plasma as Modulators of Reproduction and Fertility

**DOI:** 10.3390/ijms242417336

**Published:** 2023-12-11

**Authors:** Bernadett Nagy, Kálmán Kovács, Endre Sulyok, Ákos Várnagy, József Bódis

**Affiliations:** 1National Laboratory on Human Reproduction, 7624 Pécs, Hungary; kalman.kovacs72@gmail.com (K.K.); esulyok@t-online.hu (E.S.); varnagy.akos@pte.hu (Á.V.); bodis.jozsef@pte.hu (J.B.); 2Department of Obstetrics and Gynecology, University of Pécs, 7624 Pécs, Hungary; 3HUN-REN–PTE Human Reproduction Research Group, 7624 Pécs, Hungary; 4Medical School, University of Pécs, 7624 Pécs, Hungary; 5Doctoral School of Health Sciences, University of Pécs, 7624 Pécs, Hungary

**Keywords:** thrombocytes, reproduction, fertility, in vitro fertilization, preeclampsia, aspirin, platelet-reach plasma

## Abstract

Thrombocytes play an essential role in hemostasis and thrombosis. Moreover, the controlled activation of thrombocytes is required in reproduction and fertility. The platelet-activating factor and the controlled activation of platelets have important roles in folliculogenesis, ovulation, placental development, implantation and embryo development. Activated platelets accumulate in the follicular vessels surrounding the follicle and, due to its released soluble molecules (factors, mediators, chemokines, cytokines, neurotransmitters), locally increase oocyte maturation and hormone secretion. Furthermore, activated platelets are involved in the pathogenesis of ovarian hyperstimulation syndrome (OHSS) and preeclampsia. Low-dose aspirin can prevent OHSS during ovulation induction, while intrauterine or intraovarian administration of platelet-rich plasma (PRP) increases the endometrium thickness and receptivity as well as oocyte maturation. Activated thrombocytes rapidly release the contents of intracellular granules and have multiple adhesion molecules and receptors on their surface. Considering the numerous homeostatic endocrine functions of thrombocytes, it is reasonable to suppose a platelet-associated regulatory system (PARS) in reproduction. Although we are far from a complete understanding of the regulatory processes, the results of PARS research and the therapeutic application of aspirin and PRP during in vitro fertilization are promising.

## 1. Introduction

Several mediators are involved in reproduction and fertility, and among others, platelets also play an important role in the physiology of reproduction. It is well known that thrombocytes are a key physiological component of hemostasis, coagulation and thrombosis. It has become clear that platelets are not simply cell fragments that plug the leak in damaged blood vessels, but they have several physiological functions.

Beyond the classic function of clotting, platelets have an important role in inflammatory, neoplastic and immune processes. Thrombocytes take part in the defense action against pathogenic microorganisms at the site of invasion [1,2]. Moreover, the controlled activation of thrombocytes plays a role in reproduction and fertility [1,3]. These important cell fragments circulate within the blood with a life span of about 10 days. The activation of platelets can occur through several agonists, like thrombin, collagen, adenosine diphosphate, thromboxanes and serotonin [4]. It is proved that platelets contain several soluble molecules, factors, mediators, cytokines, chemokines and neurotransmitters like serotonin in their granules. They also have multiple adhesion molecules and receptors on their surface. These properties make platelets suitable for being active mediators or even controllers of a complex regulatory system, which, based on literature data and our own data, can form a platelet-associated regulatory system (PARS) [3].

The balance of this regulatory system is crucial to avoid over- and under-activation leading to pathologic processes. Platelets through blood clots cause strokes and heart attacks [1,5]. Platelets are also important components of the innate immune system. Expressing many immunomodulatory molecules and cytokines, they can regulate the immune system through interactions with various cells [6,7]. Uncontrolled or misdirected activation of platelets and the immune response has been implicated in the development and progression of many inflammatory conditions, such as coronary artery disease and autoimmune diseases like rheumatoid arthritis. The role of platelets in various endocrine diseases has also been reported. Hyperthyroidism and primary hyperparathyroidism are associated with increased platelet count [8,9].

A direct interaction between platelets and bacteria leading to platelet activation has been reported in several in vitro and in vivo studies [10,11]. Thrombocytes can also promote cancer. The interaction between tumor cells and platelets within the bloodstream plays an important role in the early stages of metastasis [12].

In this review, an attempt was made to summarize the physiological role of platelets by reviewing the latest literature regarding their role in reproduction (Figure 1). Platelets are important in ovarian function, endometrial thickening and receptivity, implantation as well as placentation and embryonic development. Thrombocytes also play a role in pathophysiological processes like ovarian hyperstimulation syndrome (OHSS), preeclampsia, pregnancy-induced hypertension, HELLP (hemolysis, elevated liver enzymes, Low Platelets) syndrome and intrauterine growth restriction. This review discusses the clinical significance of this knowledge and the currently available therapeutic strategies that can be applied during in vitro fertilization (IVF).

## 2. Ovarian Function Is Controlled by Platelet-Associated Regulatory System (PARS)

Bodis et al. proved that the synthesis of progesterone and estradiol in human granulosa cells is stimulated by serotonin and histamine. The question arises as to how these neurotransmitters can reach the ovary. Serotonin and histamine found in the granules of platelets reach the ovaries via the bloodstream, and platelets release these neurotransmitters after their activation [7,13,14]. Serotonin and histamine have a direct stimulating effect on ovarian steroid synthesis, so it is reasonable to suppose the paracrine–endocrine regulation of ovaries [3,15].

Serotonin increases progesterone secretion in a dose-dependent manner, while the increase in estradiol secretion is not dose-dependent [13]. The effect of histamine on granulosa cells is exerted via the H1 receptor on their surface. In the case of histamine, estradiol increases dose-dependently, while the increase in progesterone is not dose-dependent. Serotonin and histamine may play a role in the formation of the corpus hemorrhagicum at the beginning of luteinization [3].

Based on these previous findings, activated platelets stimulate ovarian steroid synthesis. Interestingly, estrogen is also able to activate platelets through platelet-activating factor (PAF), which in turn will stimulate further steroid synthesis and egg maturation. Based on animal experiments, estrogen reduces PAF-acetyl hydrolase activity, which leads to the accumulation of PAF in the ovaries [16,17]. This means that PAF synthesis may be mediated by ovarian steroid hormones during ovulation.

In preovulatory follicles, the concentration of thromboxane B2 (TXB2) in the follicular wall and follicular fluid increases significantly. TXB2 is a stable metabolite of TXA2, the source of which is platelets attached and aggregated to endothelial cells. It is assumed that platelets and the TXA2 released from them may contribute to the periovulatory process [3,7].

After ovulation, a corpus luteum is formed in the postovulatory follicle. The corpus luteum is an endocrine organ whose task is to provide progesterone. After ovulation, a significant amount of CD41 positive platelets was detected in the extravascular space around the luteinizing granulosa cells. It has been shown that platelets regulate endothelial cell migration and granulosa cell luteinization in the remodeling processes of the corpus luteum [18].

Platelets are activated by adhesion to the extracellular matrix, probably by the same mechanisms involved in hemostasis. Activated platelets express P-selectin (glycoprotein), which helps adhesion to monocytes, neutrophil granulocytes and lymphocytes. In vitro studies have shown that the progesterone production of the luteinizing granulosa cells of patients undergoing in vitro fertilization was promoted by direct contact with platelets during the 4-day culture. Platelet-derived factors increase the migration of granulosa cells and endothelial cells, which results in increased extracellular matrix formation around luteal cells and the growth of vascular networks [3].

Important elements of the interaction between platelets and ovaries are, among others, serotonin, histamine and PAF. In addition, another important molecule must be mentioned, brain-derived neurotrophic factor (BDNF), which is a growth factor derived from granulosa cells and platelets that stimulates folliculogenesis, oocyte maturation and early embryonic development [3,7,19].

## 3. Thrombocytes in Early Pregnancy

### 3.1. Thrombocytopenia as an Initial Maternal Response to Fertilization

One of the earliest responses of the mother’s body to pregnancy is a transient decrease in platelet count [7,20,21]. In mice, after the transfer of the fertilized egg, the decrease in the number of platelets correlated with the number of implanted embryos that were detectable for about 7 days. The increased consumption of platelets is one of the earliest reactions in pregnancy, which leads to a decrease in platelet concentration in the spleen and peripheral blood.

The platelet-activating factor most likely plays a role in the process [7,22]. PAF is a phospholipid activator of platelet aggregation, which is produced by white blood cells, endothelial cells and the platelets themselves. Platelets are activated by PAF during gonadotropin-induced ovulation in immature rats, and PAF is also involved in the rupture of follicles. Using the PAF antagonist, a PMSG (pregnant mare serum gonadotropin)- and hCG-induced decrease in platelet count could be prevented in animal experiments. The decrease in platelet count induced by the administration of PMSG and hCG was reversed to the level of the untreated control group by the PAF antagonist, and the inhibitory effect of PAF antagonists on ovulation and thrombocytopenia could be completely prevented by the administration of a synthetic PAF injection. When both ovaries were removed, administration of PMSG and hCG did not reduce the platelet count.

### 3.2. Early Pregnancy Factor (EPF) and Lymphocyte–Thrombocyte Adhesion

The early embryo produces substances that, when transferred to the maternal blood, increase lymphocyte–thrombocyte adhesion and induce rosette formation [23]. EPF already appears in the blood 48 h after fertilization, and its activity can be determined with the rosette inhibition test. The false negative rate for detecting pregnancy with this test is only 3.4% [24]. EPF shows a high degree of homology with the chaperone 10 protein and is characterized by the fact that it is necessary for the survival of the embryo, as well as having an immunosuppressive and growth factor effect [25].

## 4. Role of Platelets in Ovarian Hyperstimulation Syndrome

### 4.1. Platelet Activation in OHSS

Platelet activation caused by ovulation induction also plays a role in the pathomechanism of OHSS. This condition is a serious, potentially life-threatening, iatrogenic disease caused by vasoactive substances released from overstimulated ovaries [7,26]. With the increase in assisted reproduction procedures, an increase in the number of OHSS was observed. The main feature of the disease is increased capillary permeability induced by Vascular Endothelial Growth Factor (VEGF), which correlates with increased platelet activation. Activated platelets release histamine, serotonin, platelet-derived growth factor (PDGF) and lysophosphatidic acid [26,27]. These substances can cause further pathophysiological processes leading to OHSS. The exact mechanism is not clear, but several factors play a role in the development of OHSS, either indirectly or by directly acting on VEGF. Human chorionic gonadotropin (hCG) increases the expression of VEGF in human granulosa cells and also increases the serum concentration of VEGF [28,29,30]. In the case of OHSS, the increased platelet count and blood clotting factor level, as well as the associated hyperviscosity in severe forms of this syndrome, can lead to the development of intravascular thrombosis. Substances released by activated platelets contribute to the pathophysiological cascade leading to OHSS.

### 4.2. Low-Dose Aspirin Therapy to Prevent OHSS

Based on the finding that platelet activation plays a role in the pathomechanism of OHSS, it can be assumed that the use of aspirin (acetylsalicylic acid) can prevent the development of hyperstimulation syndrome [7,31]. Várnagy et al. [31] published that ovulation is an inflammatory process with significant platelet activation and that the administration of aspirin can be an effective prophylaxis of OHSS. Previous studies have reported many positive effects of aspirin during assisted reproduction treatments [32]. Low-dose aspirin treatment improves ovarian responsiveness, uterine and ovarian blood flow velocity, implantation and pregnancy rates in patients undergoing in vitro fertilization. Based on these, many IVF centers used aspirin primarily to increase the pregnancy rate. Superovulation treatment leads to platelet hyperstimulation and related OHSS, and since aspirin inhibits this process, its use should be considered for prophylactic purposes. Therefore, prophylactic aspirin therapy was included in the treatment protocol for patients receiving superovulation treatment in the IVF Center of the University of Pécs. The aim was to determine the efficacy of platelet function inhibition in OHSS prevention through a retrospective and prospective study. To prove this, aspirin was randomly administered to low- and high-risk patients for OHSS during in vitro fertilization treatment, and the frequency of OHSS was determined in each group. The high-risk group included patients with a history of OHSS, polycystic ovary syndrome and those under 30 years of age.

Among the patients treated with aspirin, only two cases (0.25%) of severe or critical stage OHSS were observed; both patients belonged to the high-risk group. However, among patients who did not receive aspirin therapy, 43 (8.4%) cases developed OHSS. At the same time, OHSS was not detected in the low-risk group. Since there was a significant difference in OHSS between aspirin-treated and aspirin-non-treated patients in the high-risk group, a conclusion can be drawn about a previously undescribed beneficial effect of aspirin. Furthermore, since a mild OHSS was observed in the group treated with aspirin, it can be assumed that aspirin is not only suitable for the prevention of the disease but also reduces the severity of the symptoms and, thus, the stage of OHSS.

With respect to the pregnancy outcome, there was no significant difference between the studied groups in the results, which is in line with other authors’ similar test results regarding aspirin [31,33]. At the same time, only two cases of severe or critical OHSS were observed with aspirin treatment. In order to prevent the disease and relieve symptoms, low-dose aspirin is recommended for patients at high risk of OHSS. With a randomized study conducted at the Department of Obstetrics and Gynecology of the University of Pécs, Hungary, it was proved that the administration of low-dose aspirin is effective in preventing severe OHSS and alleviating the symptoms.

## 5. Endometrial Effects of Platelets

The establishment of endometrial receptivity is essential for successful embryo implantation. The substances released from the platelets promote the thickening of the endometrium and increase its receptivity. Primarily, platelet-derived growth factor A (PDGFA) released from platelets can be responsible for the receptivity of the human endometrium. In the non-receptive phase of the endometrium, active PDGFA is hardly detectable, while in the receptive period, it can be detected on the apical surface of the endometrium by immunohistochemistry. Its level in the receptive period was significantly higher than in the non-receptive period, so active PDGFA can also be used as a biomarker of endometrial receptivity [7,34].

## 6. Placental Effects of Platelets

When pregnancy occurs, the controlled activation of platelets plays a role in implantation, placentation, appropriate placental vascular remodeling and maintenance of placental perfusion [3,7,21]. Bioactive mediators released from platelets, primarily IGF-1 (insulin-like growth factor) and PDGF, play a role in placentation and protect trophoblast cells from apoptosis [35]. In the case of severe thrombocytopenia, significant intrauterine growth retardation was observed in mice due to placental vascular disorder and immaturity of blood vessels [36].

Endovascular invasion of trophoblast cells into maternal spiral arteries is required for the formation of large-caliber vessels [37]. Trophoblasts invade maternal spiral arteries and transform the arteries into low-resistance large-caliber vessels. This process, required for adequate placental perfusion, is called maternal vascular remodeling. Histological examination showed deposition of maternal platelets in the trophoblast aggregates formed in the spiral arteries. In order to achieve such vascular caliber changes during early placentation, direct contact between platelets and trophoblast cells is not necessary. Soluble factors released from the activated platelets enhance the invasive capacity of isolated trophoblasts in vitro [38]. Thrombocytes attached to collagen deposited around endovascular trophoblasts are activated and expressed P-selectin that binds to P-selectin glycoprotein ligand-1 (PSGL-1) on the surface of neutrophils, leading to recruitment and activation of the neutrophils. The activated neutrophils release PAF that promotes platelet aggregation [38].

In preeclampsia, pregnancy-induced hypertension and HELLP syndrome, the hyperactivation of platelets can lead to clinical manifestation. In this case, the TXA2 and prostacyclin balance is disturbed. While in preeclampsia and pregnancy-induced hypertension, we can talk about controlled platelet hyperactivation, in HELLP syndrome, uncontrolled platelet hyperactivation and consumption cause the characteristic symptoms, hemolysis, elevated liver enzyme values and low platelet count [21]. Factors that trigger the hyperactivation of platelets and, thus, the insufficiency of the PARS regulatory system can be, for example, subendothelial collagen released during endothelial damage or simultaneous infection, which worsens the condition by causing further platelet activation. Platelet activation contributes to the thrombo-inflammation state and progression of the disorder [39].

## 7. Aspirin Treatment in Placenta-Mediated Obstetrical Complications

The inhibition of platelet aggregation with low-dose aspirin can delay or prevent preeclampsia from the first trimester onwards [40]. Low-dose aspirin has been used worldwide to prevent placenta-mediated obstetrical complications such as preeclampsia, recurrent pregnancy loss and intrauterine growth restriction [38,40]. A meta-analysis of randomized controlled trials by Roberge et al. [41] reported that when aspirin was initiated at ≤16 weeks, there was a significant reduction and a dose–response effect for the prevention of preeclampsia, severe preeclampsia and fetal growth restriction with higher dosages of aspirin being associated with greater reduction of the three outcome parameters. When aspirin was initiated at more than 16 weeks, there was only a smaller reduction of preeclampsia independent of aspirin dosage [41]. Recently published randomized studies have demonstrated the effectiveness of aspirin in the prevention of preterm birth and preeclampsia [42,43,44]. During the ASPRE (Combined Multimarker Screening and Randomized Patient Treatment with Aspirin for Evidence-Based Preeclampsia Prevention) study, high-risk pregnant women were identified using the preeclampsia screening algorithm. The algorithm used maternal serum pregnancy-associated plasma protein A (PAPP A) and placental growth factor values, as well as maternal mean arterial pressure and the average of uterine artery pulsatility values for risk estimation. During the multicenter, double-blind, placebo-controlled study, 1776 pregnant women at high risk (>1:100) for preeclampsia received 150 mg of aspirin daily from the 11–14 week of pregnancy to the 36th week. Aspirin treatment has been shown to be effective in preventing preeclampsia in premature babies. The incidence of this disease in the group treated with aspirin was only 1.6%, while in the placebo group, it was 4.3% [43]. In populations of nulliparous women, low-dose aspirin initiated between 6 weeks of gestation and 13 weeks of gestation resulted in a reduced incidence of preterm delivery before 37 weeks and reduced perinatal mortality [44]. If premature birth occurs during aspirin treatment, there may be a risk of developing persistent fetal circulation [45,46].

## 8. Platelet-Rich Plasma (PRP) Treatment

Platelet-rich plasma has become a novel treatment in reproductive medicine (Figure 2). PRP, as an autologous biologic material, minimizes the risk of immune reactions and contagious diseases [47]. However, there is a wide variation in the reported protocols for the standardization and preparation of PRP [48]. Autologous PRP is obtained through the collection of an individual’s whole blood with the addition of citrate dextrose A to prevent platelet activation prior to its use.

An initial centrifugation to remove red blood cells is followed by a second centrifugation to concentrate platelets, and then the addition of a platelet agonist activates the sample. However, there is no consensus on whether platelets must be previously activated before their application and with which agonist (calcium chloride, thrombin or collagen) [48]. PRP plays a role in tissue regeneration, angiogenesis, cell migration, differentiation and proliferation. Activated PRP releases cytokines and growth factors, like transforming growth factor-beta, fibroblast growth factor, insulin-like growth factors 1 and 2, VEGF and epidermal growth factor [49]. Besides human applications, PRP is used in animal reproductive medicine to treat endometritis and improve follicular development, oocyte competence and the uterine environment for embryo implantation [50,51,52].

### 8.1. Intraovarian Application of Platelet-Rich Plasma

Recent research reports on the effectiveness of platelet-rich plasma treatment [4,7,53,54,55,56]. PRP was injected into the ovaries guided by ultrasound or during laparoscopy. As a result of the intraovarian application of platelet-rich plasma, not only improved hormone values and antral follicle counts but also successful pregnancies have been described. Due to the high platelet concentration, the ovarian rejuvenating effect lasted for about 12 months. PRP treatment has proven to be safe and effective in early ovarian failure. The reduced ovarian reserve associated with advanced maternal age cannot be reliably treated with gonadotropin stimulation.

The mechanism of action of PRP is not exactly known, but it can be assumed that factors released from platelets, primarily growth factors (e.g., PDGF), promote the maturation of oocytes from cells present in the ovary (renewable ovarian germline stem cells or dedifferentiated cells) and regulate vessel formation [4,57]. VEGF plays a role in follicle development and regulates vascularization, while sphingosine-I-phosphate can promote follicle maturation. Another potential mechanism for improving ovarian function could be serotonin released from PRP platelets, which can stimulate the secretion of progesterone and estradiol by granulosa cells and stimulate the maturation of oocytes [54].

Autologous PRP treatment has also been successfully used in primary ovarian insufficiency. After intraovarian administration, the level of AMH (antimüllerian hormone) increased, and successful pregnancies were also reported [58].

Although the results of PRP studies are promising, there is poor standardization among research groups and clinics; however, this must be addressed soon to evaluate the efficacy of PRP treatment [4].

### 8.2. Intrauterine Administration of Platelet-Rich Plasma

Platelets play a role in the thickening and receptivity of the endometrium and, in the case of pregnancy, in the proper development of the placenta [21,34]. Recently, several studies have reported that platelet-rich plasma may stimulate endometrial growth and improve endometrial receptivity [47,59]. The infusion of PRP has anti-inflammatory properties, too. Following recurrent implantation failure (RIF), successful implantation and live birth have been reported with intrauterine administration of autologous platelet-rich plasma 24 or 48 h before embryo transfer [60,61]. Platelet-rich plasma contains many proteins, cytokines and growth factors, which have a beneficial effect on tissue regeneration, cell proliferation, angiogenesis and endometrial receptivity. The endometrial tissue contains growth factor receptors, several of which play a role in endometrial remodeling. Growth factors help endometrial receptivity, embryo implantation and embryonic development in an autocrine and paracrine way [61].

## 9. Summary and Conclusions

Platelets play a pivotal role in many key steps of reproduction. During early pregnancy, transient reductions in thrombocyte count are observed. Platelets also contribute to periovulatory events in the ovary and the subsequent changes in the corpus luteum. This review summarized the endometrial and placental effects of platelets. While controlled platelet activation is essential for many physiological processes, the hyperactivation of platelets can lead to the development of various pathologies. Clinical observations have confirmed the effectiveness of low-dose platelet cyclooxygenase inhibitors in the case of habitual miscarriage, intrauterine growth retardation and preeclampsia, which reduces platelet aggregation by inhibiting TXA2 synthesis. Low-dose aspirin is effective in preventing severe OHSS and in alleviating symptoms.

Autologous PRP treatment is a novel alternative to treat certain infertilities like thin endometrium, recurrent implantation failure and diminished ovarian reserve. Although studies suggest that PRP treatment is promising, a standardized PRP preparation protocol would be crucial to reproduce accurate results.

The granules of platelets contain soluble molecules, factors, mediators, chemokines, cytokines and neurotransmitters, and several adhesion molecules and receptors are found on their surface. Based on these properties and the fact that platelets are essential for many endocrine functions, it is reasonable to assume that a PARS may exist. Intrinsic or extrinsic triggers and/or stimuli can complement and link regulatory pathways to target tissues or cells. The signal (PAF or other tissue or cell-specific factor) originates from stimulated organs or cells (e.g., by pituitary hormones, bacteria, external factors) and activates platelets. Platelets modulate the hypothalamo-pituitary–ovarian system. The granulosa cells increase PAF secretion. The platelets activated by PAF accumulate in the follicles and their surroundings, and with the help of the factors, mediators, chemokines, cytokines and neurotransmitters are released and contribute locally to the maturation of the oocytes and the increasing production of steroid hormones. If PARS is overregulated, platelet hyperactivation and organ dysfunction can occur, leading to serious conditions like preeclampsia, HELLP syndrome or OHSS. If PARS is underregulated, platelet hypoactivation can lead to repeated implantation failure, thin endometrium or ovarian failure (Figure 3).

The relevant literature and our own results show that platelets play a fundamental role in various regulatory processes (Figure 4). Therefore, it can be assumed that platelets are active modulators of the regulatory system of reproduction. Although we are far from a complete understanding of the regulatory processes, the results of PARS research and the therapeutic application of aspirin and PRP during in vitro fertilization are promising.

## Figures and Tables

**Figure 1 ijms-24-17336-f001:**
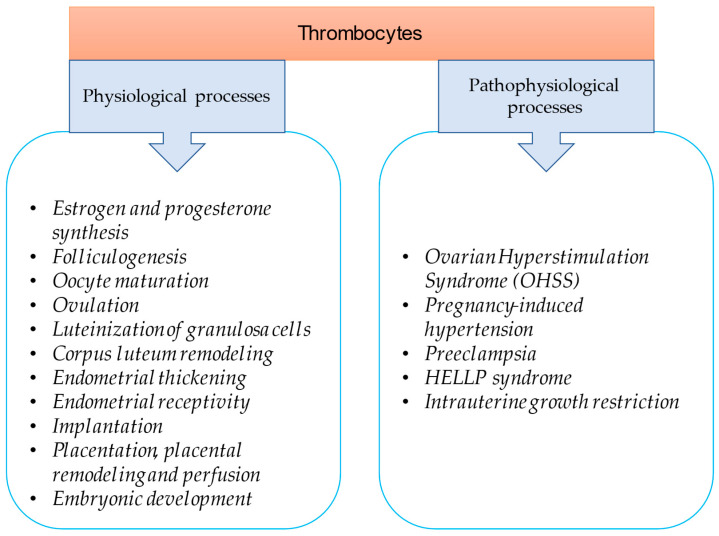
Thrombocytes as modulators of reproduction and fertility.

**Figure 2 ijms-24-17336-f002:**
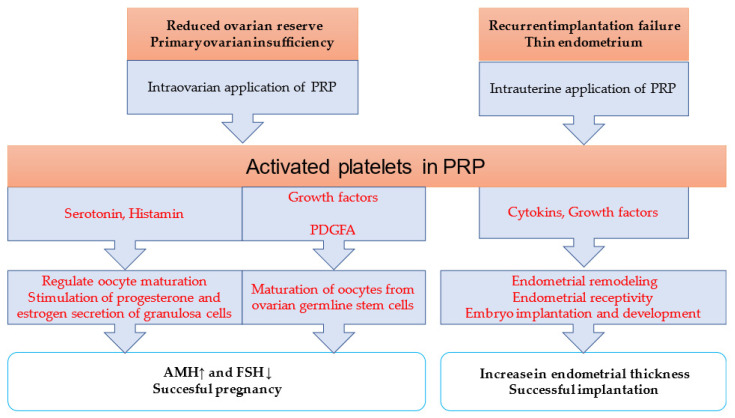
Platelet-rich plasma as a novel treatment in reproductive medicine. ↑ AMH level increase and ↓ FSH level decrease.

**Figure 3 ijms-24-17336-f003:**
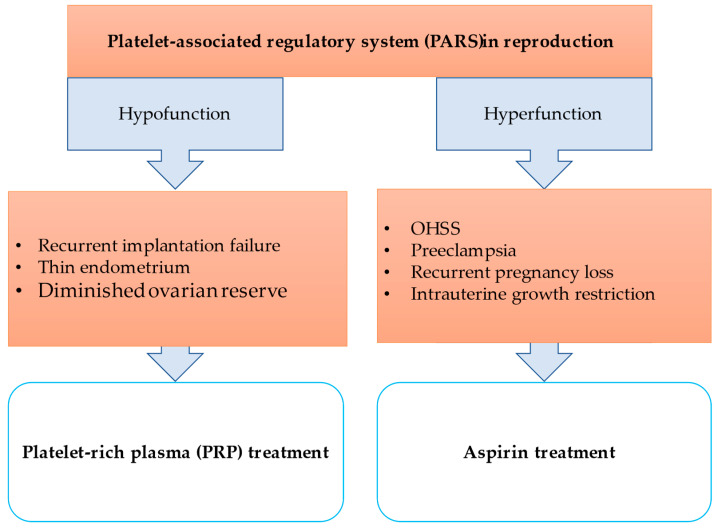
Platelet-associated regulatory system (PARS) in reproduction.

**Figure 4 ijms-24-17336-f004:**
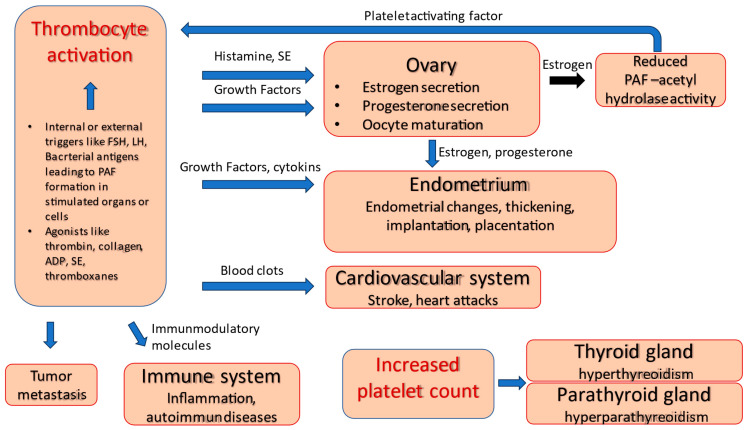
Thrombocytes as modulators of several organs.

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
