# Peer review of "Thrombocytes and Platelet-Rich Plasma as Modulators of Reproduction and Fertility"

_ijms, 2023, doi:10.3390/ijms242417336_

Round 1
Reviewer 1 Report
Comments and Suggestions for Authors
In the present work, Nagy et al. try to review thrombocytes and platelet-rich plasma (PRP) as modulators of reproduction and fertility. There are some questions that should be explained.
1. Editing of English language and style is needed. Pease revise the manuscript throughout.
For example,
Format of names of authors and references is not suitable for this Journal.
Title, ‘(PRP)’ should be deleted.
Line 63, ‘In our summary, we examine the diverse physiological role’, these should be rewritten.
‘In vitro’ and ‘In vivo’ should be italic. Pease revise these throughout the manuscript.
Line 300, ‘8.2..’
2. Figure 1 should be revised, ‘Thrombocytes as Modulators of Reproduction and Fertility’ should be deleted in Figure. Some physiological processes are repeated. Some explains for Figure 1 should been added.
3. As a Review article, Discussion section is not suitable. The whole construction of this Review article should be revised. In addition, at least, one representative figure, including some organs, should be added.
4. There are so many new references are related to this manuscript, but this manuscript does not cite them. For example,
Atkinson L, Martin F, Sturmey RG. Intraovarian injection of platelet-rich plasma in assisted reproduction: too much too soon? Hum Reprod. 2021;36(7):1737-1750.
de Almeida LGN, Young D, Chow L, Nicholas J, Lee A, Poon MC, Dufour A, Agbani EO. Proteomics and Metabolomics Profiling of Platelets and Plasma Mediators of Thrombo-Inflammation in Gestational Hypertension and Preeclampsia. Cells. 2022;11(8):1256.
Segabinazzi LGTM, Canisso IF, Podico G, Cunha LL, Novello G, Rosser MF, Loux SC, Lima FS, Alvarenga MA. Intrauterine Blood Plasma Platelet-Therapy Mitigates Persistent Breeding-Induced Endometritis, Reduces Uterine Infections, and Improves Embryo Recovery in Mares. Antibiotics. 2021;10(5):490.
Ghallab RS, El-Beskawy M, El-Shereif AA, Rashad AMA, Elbehiry MA. Impact of intrauterine infusion of Platelets-Rich plasma on endometritis and reproductive performance of Arabian mare. Reprod Domest Anim. 2023;58(5):622-629.
Lin Y, Qi J, Sun Y. Platelet-Rich Plasma as a Potential New Strategy in the Endometrium Treatment in Assisted Reproductive Technology. Front Endocrinol. 2021;12:707584..
Mouanness M, Ali-Bynom S, Jackman J, Seckin S, Merhi Z. Use of Intra-uterine Injection of Platelet-rich Plasma (PRP) for Endometrial Receptivity and Thickness: a Literature Review of the Mechanisms of Action. Reprod Sci. 2021;28(6):1659-1670.
Borş SI, Ibănescu I, Borş A, Abdoon ASS. Platelet-rich plasma in animal reproductive medicine: Prospective and applications. Reprod Domest Anim. 2022;57(11):1287-1294.
Barad DH, Albertini DF, Molinari E, Gleicher N. Preliminary report of intraovarian injections of autologous platelet-rich plasma (PRP) in extremely poor prognosis patients with only oocyte donation as alternative: a prospective cohort study. Hum Reprod Open. 2022;2022(3):hoac027.
Comments on the Quality of English LanguageModerate editing of English language required.
Reviewer 2 Report
Comments and Suggestions for Authors
Comments about the manuscript:
“Thrombocytes and Platelet-rich Plasma (PRP) as Modulators of Reproduction and Fertility”
The activation of thrombocytes, known for their involvement in hemostasis and thrombosis, is necessary for reproduction and fertility. (folliculogenesis, ovulation, implantation, placentation, embryonic development). Activated platelets are also involved in preeclampsia and ovarian hyperstimulation syndrome (OHSS) which can be prevented by low-dose aspirin. Intrauterine or intraovarian administration of platelet-rich plasma (PRP) increases endometrial thickness and receptivity and oocyte maturation. The aim of this review was to highlight the effects of thrombocytes in reproductive physiology and to share therapeutic strategies linked to currently available platelets that can be applied during in vitro fertilization.
This review seems useful and well done to me. I have just a few minor comments to make for improving the manuscript.
General: given the large number of abbreviations, a list of all the abbreviations could be placed at the end of the text: this would be very useful for the reader.
Page 6, line 243. “Roberge et al.”: add a numbered reference, and add it to the list if necessary.
Page 6, line 247. Replace “at >16 weeks” with “at more than 16 weeks”.
Page 6, line 268. “ preparation of PRP (52)”.
- The last reference is [44]: there is no reference between [44] and [52]?
- References [45] to [48] are on the next page, after [53].
- References must be checked and eliminated in numerical order.
Page 6, line 274. “which agonist (calcium chloride, thrombin, or collagen) (52, 53).”: there is no reference [53] in the list.
Page 8, line 355: write “Figure 3” between brackets.
Check reference numbers and use square brackets [ ] instead of brackets ( ).
Round 2
Reviewer 1 Report
Comments and Suggestions for Authors
Thanks for author’s responses. However, I do not think this manuscript is suitable for publication in this journal at present.
1. Editing of English language and style is needed. Pease revise the manuscript throughout.
For example,
Lines 37, 72, ‘(Figure 1.)’ and ‘(Figure 2.)’, should be changed to ‘(Figure 1)’ and ‘(Figure 2)’. In addition, ‘Figure 3.’ and ‘Figure 3.’.
The titles for Figures 1 and 2, some word should be not capitalized.
There are so many explanation for abbreviation in the text. ‘Early Pregnancy Factor (EPF)’, Ovarian hyperstimulation syndrome (OHSS)…..
Some paragraphs are only two or three sentences. These should be revised. Lines 294-295.
2. Figure 1, a conclusive Figure for this review paper may be present at the end.
3. In general, scientific papers are written in the third-person manner rather than the first person. Please check this throughout the manuscript. There are so many ‘we’ and ‘our’.
4. Lines 174-177, ‘Our study on OHSS was based on our…..’; Lines 185-190, ‘Our goal was to determine the efficacy…..; Lines 191-194, ‘In our study, among the patients treated with aspirin,…’. These sentences should be revised.
5. As a review manuscript, there are too citation in the ‘Summary and conclusion’ section. In addition, ‘(Aspirin®)’ should be not present in the ‘Summary and conclusion’ section.
6. Format of references is not suitable for this Journal.
Comments on the Quality of English LanguageModerate editing of English language required.
Round 3
Reviewer 1 Report
Comments and Suggestions for Authors
Thanks for author’s responses. However, Summary and conclusion section is too long, and it should be refined.
Comments on the Quality of English LanguageMinor editing of English language required.
Author Response
Response to reviewer 1 round 3:
Thank you so much for your supportive comments.
Thanks for author’s responses. However, Summary and conclusion section is too long, and it should be refined.
Minor editing of English language required.
With the help of a native speaker lector we made minor editing of English language (Line 327, 328, 329, 340, 346) and we made a significant reduction of the Summary and conclusion section (403 word instead of 537 word).
We deleted the following phrases and sentences:
’The above results prove that platelets’
’This review discussed the phenomenon of the’
, which may even be suitable for early detection of pregnancy
’ (e.g. OHSS, preeclampsia, HELLP).’
’The administration of’
’ (<7 mm), ’
’in reproductive medicine’
’consistent and’
’Hypothalamic GnRH releases FSH from the anterior pituitary, which induces and stimulates the maturation of follicles and oocytes and the secretion of steroid hormones in the ovary. At the same time, the’
’in the follicles’
’As a result of these processes, the’
’The activation of platelets means their aggregation and sludge formation, as well as the release of the mentioned biologically active factors, which can locally amplify tissue-specific cell reactions. If this process is damaged or inhibited for any reason, then the targeted stimulated organ shows dysfunction’
’Based on the data summarized above’,
’complex’
Thank you very much for reviewing our manuscript.
We look forward to your favourable response.
Yours sincerely,
Bernadett Nagy.